META-RESEARCH ARTICLE

# Shortcut citations in the methods section: Frequency, problems, and strategies for responsible reuse

Kai Standvoss[1,2,3‡], Vartan Kazezian[4‡], Britta R. Lewke[5], Kathleen Bastian[6], Shambhavi Chidambaram[7], Subhi Arafat[3,8], Ubai Alsharif[9], Ana Herrera-Melendez[10], Anna-Delia Knipper[11], Bruna M. S. Seco[12¤], Nina Nitzan Soto[13], Orestis Rakitzis[14], Isa Steinecker[2,15], Philipp van Kronenberg Till[16], Fereshteh Zarebidaki[17], Parya Abbasi[4], Tracey L. Weissgerber[4]*

1 Department of Psychiatry, Charité–Universitätsmedizin Berlin, Corporate Member of Freie Universität Berlin and Humboldt-Universität zu Berlin, Berlin, Germany, 2 Bernstein Center for Computational Neuroscience, Charité—Universitätsmedizin Berlin, Berlin, Germany, 3 Einstein Center for Neurosciences Berlin, Berlin, Germany, 4 QUEST Center for Responsible Research, Berlin Institute of Health at Charité–Universitätsmedizin–Berlin, Berlin, Germany, 5 Berlin School of Mind and Brain, Humboldt-Universität zu Berlin, Berlin, Germany, 6 CERPOP-UMR1295, Université de Toulouse III, Toulouse, France, 7 Department of Life Sciences, Humboldt-Universität zu Berlin, Berlin, Germany; Berlin School of Mind and Brain, Berlin, Germany, 8 Department of Neurology, Charité—Universitätsmedizin Berlin, Berlin, Germany, 9 Department of Oral and Maxillofacial Surgery, Dortmund General Hospital, Dortmund, Germany, 10 Department of Psychiatry and Psychotherapy, Campus Benjamin Franklin, Charité–Universitätsmedizin Berlin, Corporate Member of Freie Universität Berlin and Humboldt-Universität zu Berlin, Berlin, Germany, 11 German Federal Institute for Risk Assessment (BfR), Department of Biological Safety, Berlin, Germany, 12 Former affiliation: Department of Biomolecular Systems, Max Planck Institute of Colloids and Interfaces, Potsdam, Germany, 13 Institute du Cerveau–Paris brain institute (ICM), Paris, France; École doctorale Cerveau, cognition, comportement (ED3C), Sorbonne Université Paris, Paris, France, 14 Department of Psychiatry and Neurosciences, Charité–Universitätsmedizin Berlin, Corporate Member of Freie Universität Berlin and Humboldt-Universität zu Berlin, Berlin, Germany, 15 Biological Psychology and Neuroergonomics, Berlin Institute of Technology, Berlin, Germany, 16 Berlin Institute of Health and Neuroscience Research Center, Charité–Universitätsmedizin Berlin, Berlin, Germany, 17 Former affiliation: Institute of Neurophysiology, Charité—Universitätsmedizin, Berlin, Germany

¤ Current address: Biontech SE, Mainz, Germany
‡ These authors share first authorship on this work.
* tracey.weissgerber@bih-charite.de

**Data Availability Statement:** The abstraction protocols, data and code for the methodological citations study and journal policy study were

## Abstract

Methods sections are often missing essential details. Methodological shortcut citations, in which authors cite previous papers instead of describing the method in detail, may contribute to this problem. This meta-research study used 3 approaches to examine shortcut citation use in neuroscience, biology, and psychiatry. First, we assessed current practices in more than 750 papers. More than 90% of papers used shortcut citations. Other common reasons for using citations in the methods included giving credit or specifying what was used (who or what citation) and providing context or a justification (why citation). Next, we reviewed 15 papers to determine what can happen when readers follow shortcut citations to find methodological details. While shortcut citations can be used effectively, they can also deprive readers of essential methodological details. Problems encountered included difficulty identifying or accessing the cited materials, missing or insufficient descriptions of the cited method, and shortcut citation chains. Third, we examined journal policies. Fewer than

deposited on the Open Science Framework (RRID: SCR_003238) at https://osf.io/d2sa3/.

**Funding:** This study was completed through a participant guided, learn-by doing meta-research course funded by the Berlin University Alliance within the Excellence Strategy of the federal and state governments (301_TrainIndik to TLW). VK received salary support from the Berlin University Alliance grant. The funders had no role in study design, data collection and analysis, decision to publish, or preparation of the manuscript.

**Competing interests:** The authors have declared that no competing interests exist.

one quarter of journals had policies describing how authors should report previously described methods. We propose that methodological shortcut citations should meet 3 criteria; cited resources should provide (1) a detailed description of (2) the method used by the citing authors', and (3) be open access. Resources that do not meet these criteria should be cited to give credit, but not as shortcut citations. We outline actions that authors and journals can take to use shortcut citations responsibly, while fostering a culture of open and reproducible methods reporting.

## Introduction

Methods sections should serve several purposes, each of which requires a different level of detail (Fig 1). Well-written methods sections provide an overview of the study design and techniques used to answer the research question, help readers to evaluate the risk of bias and provide details needed to replicate the experiment. Unfortunately, methods sections are often missing critical details. The Reproducibility Project: Cancer Biology aimed to replicate high profile findings from 193 experiments in cancer research [1]. Unfortunately, none of the papers contained sufficient details to allow researchers to design and conduct a replication study [2]. When the original authors were contacted to obtain methodological details, 41% were extremely or very helpful, 9% were minimally helpful, and 32% were not helpful or did not respond [2]. An assessment of 300 fMRI studies revealed that key information, such as how the task was optimized for efficiency or the distribution of inter-trial intervals, was frequently missing [3]. Methodological

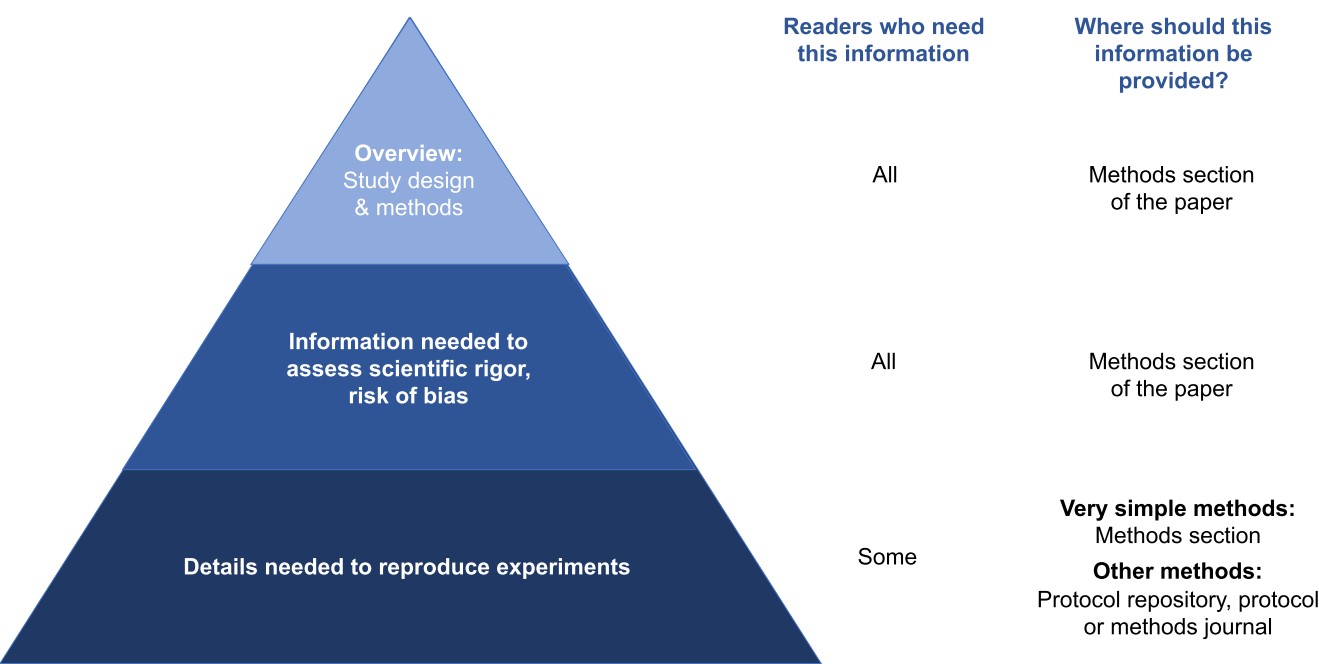

**Fig 1. The level of methodological detail required depends on the reader.** All readers need an overview of the study design, methods used to answer the research question, and information needed to assess scientific rigor and the risk of bias. These details should always be presented in the methods section of the paper. While fewer readers need the details required to reproduce or reuse the method, these individuals are particularly important because they are most likely to perform follow-up experiments. Very simple methods that can be explained and reproduced easily may be described in the methods section. Protocol repositories or method or protocol journals are better for many methods, as it is difficult to provide the details needed to implement or reuse the method within the methods section.

details in randomized controlled trials are also frequently missing. In approximately half of randomized controlled trials included in Cochrane reviews, methods were not reported in enough detail to allow researchers to complete risk of bias assessments for items such as randomization sequence generation and allocation concealment [4].

A methodological shortcut citation is a citation that is used to replace a full description of a method or a part of a method, under the assumption that the resource being cited fully describes the method. Shortcut citations are sometimes accompanied by phrases like "as described previously" or "see (citation) for details." Methodological shortcut citations are one factor that could affect readers' ability to reproduce an experiment.

While some scientists and editors view the use of shortcut citations as a good practice [5,6], others worry that these shortcuts may adversely affect reproducibility [7]. Ideally, shortcut citations should reference resources that describe the method, as it is currently performed, in detail. This may include protocol papers, diagnostic guidelines, or original research articles with detailed methods. In practice, anecdotal reports show that shortcut citations can cause problems [8,9]. Readers may be unable to access the cited paper. The cited paper may not contain methodological details or may itself use a shortcut citation instead of describing the methods. Those in favor of shortcut citations note that authors do not waste time repeating details that have been written elsewhere and avoid potential copyright issues that might emerge if one copied methods from another publication. Those who are skeptical of shortcuts argue that the cited paper may no longer reflect current practice or may not accurately describe the procedures of the authors who cite it. Furthermore, consulting cited resources to obtain details needed to interpret the study is time consuming. There are also disagreements about how shortcut citations should be used. While some authors cite the paper that introduced the method to give its creators credit, others cite the paper whose methods most closely resemble their own. Some journals require authors to use shortcut citations to avoid repeating published methods, or incentivize this practice through strict word limits, whereas other journals have excluded the methods section from word limits to encourage detailed reporting [10].

Our meta-research study used 3 approaches to systematically examine the use of methodological shortcut citations in neuroscience, biology, and psychiatry. First, we examined papers to determine why authors use citations in the methods section and to assess how often shortcut citations were used. Next, we reviewed shortcut citations for 15 papers to determine what can happen when readers follow these citations to find methodological details. Third, we reviewed journal policies related to shortcut citations and methodological reporting.

## Methods

This study was performed as part of a participant guided, learn-by-doing course [11], in which graduate students in different fields at 4 Berlin universities learned meta-research skills by working together to design, conduct, and publish a meta-research study.

We conducted 3 distinct but related studies.

1. **Methodological citations study:** This examined the reasons why authors use citations in the methods section of papers and assessed how often shortcut citations were used. Additional data collected included the number of resources cited in a shortcut citation and the years in which papers cited as shortcuts were published.

2. **Shortcut citation chains study:** This study examined problems that may occur when readers consult shortcut citations to find further details of the study methods.

3. **Journal policy study:** This study examined journal policies related to shortcut citations and methodological reporting.

Each study examined 3 fields: neuroscience, biology, and psychiatry, to improve generalizability. These fields were selected based on the study teams' expertise. The abstraction protocols, data, and code for the methodological citations study and journal policy studies were deposited on the Open Science Framework (RRID:SCR_003238) at https://osf.io/d2sa3/ [12].

## Methodological citations study

This was a cross-sectional study of research articles.

**Systematic review.**    We followed all relevant items in the PRISMA guidelines [13]. Items that only applied to meta-analyses or were not relevant to literature surveys were not followed. Ethical approval was not required.

**Journal screening.**    Our sampling frame included journals with the highest impact factors that publish original research in each field. Journals for each category were ranked according to 2019 impact factors listed for the specified categories in Journal Citation Reports. We excluded journals that did not publish original research or did not publish a March 2020 issue. Fewer journals were used for biology ($n = 15$), compared to neuroscience and psychiatry ($n = 20$), due to the large number of publications in some biology journals. The neuroscience journals examined generally focus on biomedical neuroscience, which may include basic science, translational and clinical research (S1 Table). The biology journals examined publish a wide range of general biology research and are not exclusively biomedical (S2 Table).

**Search strategy.**    Articles were identified through a PubMed search. We performed a supplemental Web of Science search to identify articles published in journals that were not indexed in PubMed. The full search strategy is available on the OSF repository [12].

**Inclusion and exclusion criteria.**    The study included all full-length, original research articles that included a methods section published in each included journal between March 1 and March 31, 2020. Among journals that publish print issues, we examined all articles included in print issues of the journal that were published in March 2020 (S1–S3 Tables and S1 Fig). Articles for online journals that did not publish print issues were included if the publication date was between March 1 and March 31, 2020. Articles were excluded if they were not full-length original research articles, did not contain methods sections, or were methods articles.

**Screening.**    Screening for each article was performed by 2 independent reviewers (Biology: PvKT, SC, ADK, VK; Neuroscience: KS, FZB, SA; Psychiatry: IS, OR, UA) using Rayyan software (RRID:SCR_017584), and disagreements were resolved by consensus discussions between the 2 reviewers. A list of articles was uploaded into Rayyan. Reviewers independently examined each article and marked whether the article was included or excluded.

**Abstraction.**    All abstractors completed a training set of 35 articles before abstracting data. Data abstraction for each article was performed by 2 independent reviewers (Biology: SC, ADK, VK, PvKT; Neuroscience: KS, FZB, SA; Psychiatry: BL, IS, AHM). When disagreements could not be resolved by consensus, ratings were assigned after a group review of the paper. Eligible manuscripts were reviewed in detail to evaluate the following questions according to a predefined protocol (available at: https://osf.io/d2sa3/ [12]).

The following data were abstracted:

1. Is the paper related to the SARS-CoV-2 pandemic? This information was abstracted as, at the beginning of the pandemic, members of the scientific community was concerned about the quality of COVID-19 papers due to the speed at which these studies were conducted and published.

2. Does the paper include additional methodological details in the supplement?

3. Does the paper reference a repository or repositories as a shortcut for *any* method used? If so, what is the name of the repository? (Note that code used to analyze data was considered a method.)

Abstractors then reviewed each citation in the methods section, including citations listed in STAR (Structure Transparent Accessible Reporting) methods tables [14]. Some journals use these tables to provide an overview of all of the key reagents used in the study. Notations or hyperlinks that appeared only in the text, and not as entries in the reference list (e.g., company names), were not counted as citations. Links to supplemental files were not counted as citations. Multiple papers cited as part of the same reference were treated as a single citation (e.g., 2 citations appearing within the same set of brackets). Papers cited in different locations in the same sentence were treated as different citations. Citations in the supplemental methods were not evaluated.

Each methodological citation was classified into one of the categories outlined in Table 1, according to the inferred purpose of the citation. The "How (Explain a method)" and "Other"

**Table 1. Reasons for citations in the methods section.**

| Category | Example | Potential shortcut |
|---|---|---|
| **How (Explain a method):** The citation was intended to explain how something was done. This category included 4 subcategories for specific types of citations (1. General, 2. Protocol, 3. Prior publication describing the study design and/or results, and 4. Guideline or manual). | Liver samples were collected, sectioned, and frozen as described previously (citation). | Yes |
| **Who or what (Give credit or specify what was used):** The citation is used to give credit to the group that created the method, tool, resource or substance, or to specify exactly what method, tool, resource, or substance was used. The citation is not used to explain how the method was performed, or how the tool, resource or substance was used. This category includes 3 subcategories (1. Software, 2. Atlas, 3. Other). | Analyses were conducted using R (citation of R). | No |
| **From where (Source of materials):** The citation is used to show where a substance or organism was obtained from, not how the substance or organism was created. | Mice were obtained from the Smith lab (citation). | No |
| **Why (Provide context or a justification):** The citation refers to previous studies to provide context. This includes citations that compare the authors' results with results from previous studies, demonstrate that others have used a similar approach, or explain why the authors chose a particular method, model, formula, etc. The citation is not intended to provide insight into how the existing study was conducted. | In accordance with previous studies (citation), we observed that the optimal treatment dose was 20 ml. | No |
| **Formula or value:** The citation referred to a formula or specified a value for a parameter that was used. The formula or value had to be stated in the article, so that the reader would not need to look up this information in the cited resource. The citation was not used to explain how that parameter was calculated, how the formula was derived, or why the parameter or formula was chosen. | Drug cost $X for drug Y was obtained from publicly available reports (citation). | No |
| **Other:** Citations that do not fit into the other categories. | | Yes |

A complete protocol with examples is available in the online repository.

categories could be shortcut citations, whereas other categories could not. Each citation in these 2 categories was classified as a probable shortcut, possible shortcut, or no shortcut. The conceptual goal was to distinguish between situations where readers would likely need to consult the cited paper to implement the method (probable shortcut), situations where the reader may need to consult the shortcut citation to implement the method (possible shortcut), and cases where the reader would not need to consult the shortcut citation to implement the method. However, these conceptual definitions are highly subjective and depend on the reader's knowledge of the reported methods. We therefore used the syntactic definitions described below to classify shortcut citations.

○ **Probable shortcut:** The sentence that includes the shortcut citation is the only description of the method. Additional details are not provided in the following sentences or elsewhere in the methods section.

○ **Possible shortcut:** Additional details of the cited method are provided in the sentences following the sentence containing the shortcut citation, or elsewhere in the methods section.

○ **Not a shortcut:** A reader would not need to consult the cited paper to implement the method. This rare category was generally used when the citation referred to concentrations, parameters, or other details that were fully specified in the sentence where the shortcut citation appeared.

Two independent reviewers (SA and UA) abstracted the following data for each paper:

- The minimum and maximum number of resources cited within each probable shortcut.

- The minimum and maximum number of resources cited within each possible shortcut.

- The publication year of the youngest and oldest probable shortcut citations.

- The publication year of the youngest and oldest possible shortcut citations.

**Protocol modification.**   During peer review, a reviewer requested that we add publication years for all resources cited as shortcuts, rather than using the minimum and maximum to quantify the range of values observed. This data was abstracted by a single reviewer (PA).

## Shortcut citation chains study

**Selection of articles.**   Detailed assessments were performed on a cohort of studies. In each field, papers from the methodological citations study were divided into quintiles based on the total number of shortcuts (possible + probable). Ten articles from each quintile were randomly selected using a computer algorithm and placed on an ordered list. Potential reviewers identified articles on the list that fell within their area of expertise. We then chose the first article in each list that had a self-identified expert evaluator. This approach was used to select 15 parent articles (1 article per quintile per field).

**Data abstraction.**   Each reviewer carefully examined all papers or materials cited as shortcuts to determine whether they could locate the cited content. While articles were almost always accessible, reviewers were sometimes unable to access other types of cited resources. Books were only checked if they could be accessed online, as students could not visit libraries due to COVID-19 pandemic restrictions. Newer versions of books and manuals were reviewed if older, cited versions were unavailable or inaccessible.

For each cited article or resource that was found, the reviewer documented the cited material type (paper, protocol, book, etc.), publication year, whether the article or resource was open access or behind a paywall, and any other problems encountered while searching for

information on the cited method. Additionally, reviewers noted whether book citations included chapters or pages. Finally, reviewers also noted whether the article or resource contained an adequate description of the method cited in the parent article. Descriptions might be judged as inadequate if they were clearly missing essential details that would be needed to implement the methods. If the methodological description was not adequate and the shortcut citation also used a shortcut citation to explain the method cited in the parent paper, then the reviewer repeated the abstraction process for each new shortcut citation, adding these new shortcuts as additional steps in the shortcut citation chain. Abstraction was complete when the reviewer either found a complete and comprehensive description of the method or reached a dead-end in the chain of shortcut citations. Dead ends included an inability to locate the cited article or resource, or an article or resource that did not describe the cited method.

A second reviewer assessed the accuracy of all abstracted information. A graphic illustrating the number of shortcuts and chains of shortcut citations was created for each parent paper.

## Journal policy study

In this cross-sectional analysis of journal policies, we examined policies of all eligible journals listed in the Journal Citation Reports 2019 ranking for neuroscience, psychiatry, and biology.

**Inclusion and exclusion criteria.** Journals that publish original research were included, whereas journals that only publish review articles, book series, correspondence, perspectives, or editorials were excluded. Methods journals were excluded, as these journals often require authors to report extensive methodological details. Journals were also excluded if they did not have a website, had suspended publishing activities, or planned to cease publishing in 2021.

**Search strategy.** Journal webpages were examined to confirm that journal policies were accessible. Electronic searches were performed using the terms "[journal name]," "journal citation reports ranking," "author guidelines," "journal policy," and "impact factor." When journals with similar names were identified, impact factors were used to confirm that the correct journal was selected.

**Screening.** Each journal was screened by 2 of the 3 independent reviewers (BMSS, KB, NNS) to determine whether the journal was eligible according to the prespecified criteria listed above. Discrepancies were resolved by consensus. Information on whether the journal publishes original research articles was assessed through the journal descriptions (e.g., "About the Journal" or "Aims and Scope"), or by determining whether the submission guidelines listed original research as an article type. If this information was inconclusive, the 2 most recent issues were examined to determine whether the journal published original research.

**Abstraction.** Training was performed on the 20 eligible journals with the highest impact factor for each category prior to data abstraction. Data were collected by 2 of the 3 independent abstractors (BMSS, KB, NNS). When disagreements could not be resolved by consensus, discrepancies were resolved through discussion with the third reviewer. An additional study team member (TLW) was consulted to resolve complex discrepancies. A trained abstractor abstracted data from non-English journal webpages with the help of a native speaker.

Webpages containing author and or submission guidelines were identified by looking for the following terms in the website menu, or by using the website search function: "policy," "policies," "author," "author/s guidelines," "author/s instructions," "submission guidelines," "submit your article," "recommendation," "about," "publish." Each webpage section was visually examined using the search terms "method," "methods," "experiment," "reproducibility," "replicate," "replication," "repository," "repositories," "self-plagiarism," "supplementary," "supplementaries," "citation," "protocol," "journal protocol," "scientific society," and "society."

Reviewers determined whether the instructions for authors or journal policies addressed the following points:

1. Explicitly asked authors to provide sufficient methodological details to allow others to reproduce the experiment.

2. Specified how authors should describe methods that have been reported elsewhere (e.g., provide a citation instead of describing the method, briefly summarize the method and provide a citation).

3. Encouraged authors to use supplemental files, protocol repositories, or protocol journals to explain their methods.

Reviewers assessed whether this information was found in the material and methods section of the author guidelines or journal policies, in other sections of the author guidelines, or elsewhere on the journal's website.

Reviewers also consulted the journal website to determine whether the journal was affiliated with a scientific society, and whether the journal endorsed the TOP (Transparency and Openness Promotion) guidelines. The Center for Open Science list of journals that have implemented the TOP guidelines (https://www.cos.io/initiatives/top-guidelines, https://osf.io/mwxb3/) was consulted to identify journals that endorsed TOP, but did not provide this information on their website.

## Data analysis

Code for data figures and color schemes was adapted from a previous paper [15]. Summary statistics were calculated using Python (RRID:SCR_008394, version 3.7.7, libraries NumPy 1.18.5 [16], Pandas 1.2.4 [17] and Matplotlib 3.4.1 [18,19]). Charts were prepared with a Python-based Jupyter Notebook (Jupyter-client, RRID:SCR_018413 [20], Python version 3.7.7, libraries NumPy 1.18.5 [16], Pandas 1.2.4 [17] and Matplotlib 3.4.1 [18,19]) and assembled into figures with vector graphic software.

## Results

### Methodological citations study

**Study sample.** The study sample consisted of 224 articles with 2,756 methodological citations in neuroscience, 431 papers with 6,226 methodological citations in biology, and 160 papers with 1,870 methodological citations in psychiatry (Figs 2A and S1). The sample contained few publications related to COVID-19 (neuroscience: $n = 0$, 0%, biology: $n = 2$, 0.5%, psychiatry: $n = 0$, 0%). While it was not feasible to collect data on the types of methods used, the study team's subjective impression is that the papers examined covered an extensive range of methods used in each field. The biology journals published both biomedical and non-biomedical research, including field studies, computational research, and laboratory studies. Neuroscience and psychiatry journals had a strong biomedical focus. Psychiatry research is predominantly clinical, and resources cited as shortcuts often included diagnostic guidelines and surveys. Neuroscience included basic, translational, and clinical studies.

**Use of citations in the methods section.** Citations were common in the methods sections of published papers (Fig 2B, left panel). The median number of citations in the methods section was 10 [interquartile range: 6, 16.25] in neuroscience, 12 [6, 19] in biology, and 10 [7, 16] in psychiatry.

In neuroscience and psychiatry, 53% to 54% of citations in the methods section were used to explain study methods (How citations, Fig 2A). Citations were also commonly used to give

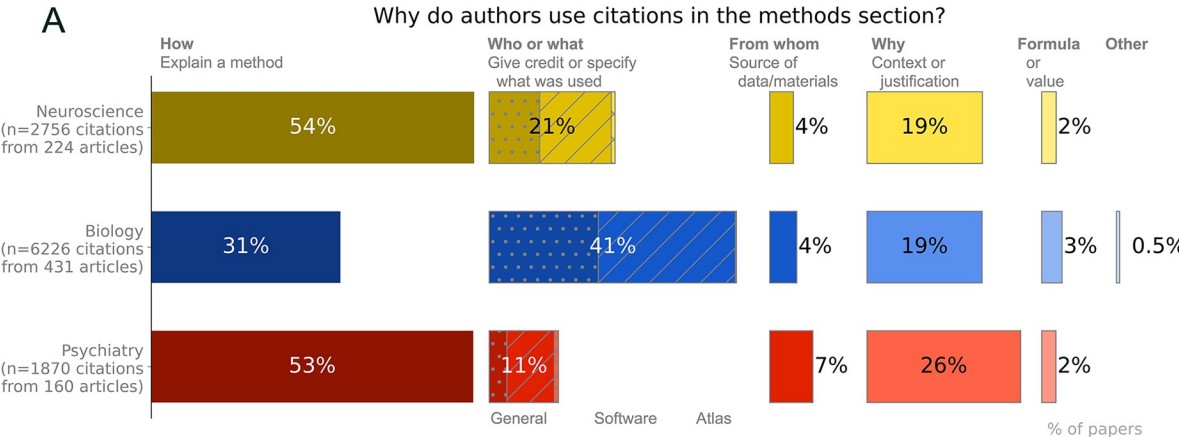

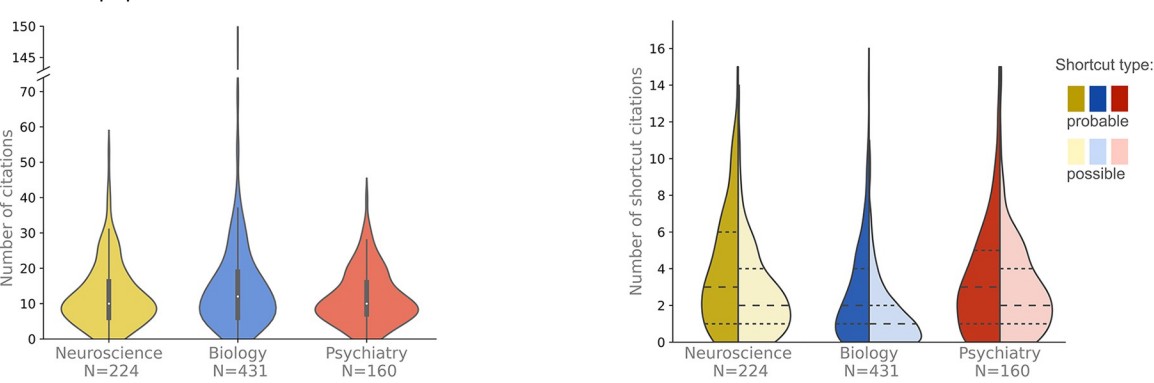

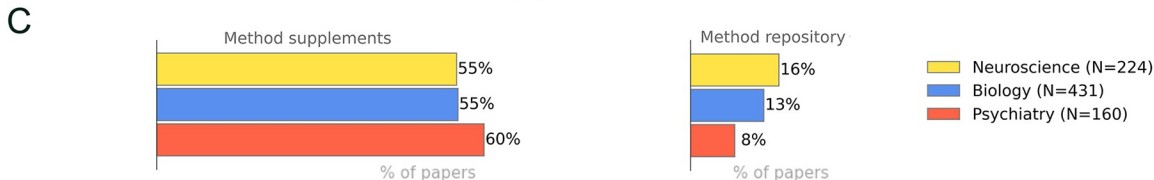

**Fig 2. Understanding the use of citations in the methods sections, methods supplements, and methods repositories.** (A) The most common reasons why authors cite papers in the methods section are to explain how a method was performed, give credit or specify what was used (who or what), or provide context or a justification (why). These numbers should be regarded as approximations. Small changes in the wording or position of the reference could alter the categorization, as could variations in reader expertise (see limitations). (B) Citations and shortcut citations often appear in the methods section of published papers. In the violin plot on the right, values for probable shortcut citations are shown in darker hues on the left half of each violin, whereas possible shortcut citations are shown in lighter hues on the right side of each violin. Most papers contain both probable and possible shortcut citations. Dashed lines on each half of the violin indicate the 25th, 50th, and 75th percentiles. Median (25th percentile, 75th percentile) for the number of probable shortcut citations per paper were as follows: neuroscience 3 (1, 6); biology 2 (1, 4); psychiatry 3 (1, 5). Median and 25th and 75th percentiles for the number of possible shortcut citations per paper were as follows: neuroscience 2 (1, 4); biology 1 (0, 2); psychiatry 2 (1, 4). (C) Methods are often shared in supplements but are less likely to be deposited on methods repositories. Percentages may not total 100% due to rounding errors. Data are available at https://osf.io/d2sa3/, in the methodological citations study folder [12].

credit or specify what was used (Who or what, 11% to 21% of papers), and provide context or a justification (Why, 19% to 26%). In biology, the most common reason for citing a paper in the methods section was to give credit or specify what was used (Who or what, 41%), followed by explaining a method (How, 31%), and providing context or a justification (Why, 19%). Citations specifying the source of data or materials (From where), or referring to a formula or

value, were uncommon in all 3 fields. Citations that did not fit into any of these categories were rare.

Depending on the field, 55% and 60% of papers provided some methodological information in supplemental files, whereas only 8% to 16% provided methodological information in a repository (Fig 2C). While reviewers did not systematically collect data on the content or quality of information in the supplemental methods, reviewers' subjective impression was that supplemental methods were rarely detailed. Common examples of supplemental methods included tables that list primers or sequences or provided basic information on study participants. Methodological information in repositories included a mixture of study protocols and code for data analysis. The most common repository was GitHub, followed by ClinicalTrials.gov, OSF and FigShare (S4 Table). Other repositories were rarely used in this dataset.

**Shortcut citations.** Methodological shortcut citations were common in all 3 fields, with 96% of neuroscience papers, 90% of biology papers and 92% of psychiatry papers containing at least 1 possible or probable shortcut. Fig 2B (right panel) shows the median number of possible and probable shortcuts for each field.

Reviewers assessed the age of all resources cited as shortcuts. Fig 3A shows the number of probable (left side of violin plot) and possible (right side of violin plot) shortcut citations in each field. The median age of the youngest shortcuts citation ranged between 3 and 5 years, whereas the median age for the median shortcut citation ranged between 6 and 11 years. Median age for the oldest shortcut citations ranged between 9 and 24 years. Summary statistics for Fig 3A are reported in S5 Table.

When using a shortcut citation, authors typically cite 1 resource (Fig 3) Shortcut citations citing 2 resources are also common, whereas citations of 3 or more resources are less common.

## Shortcut citation chains study

Figs 4 and 5 show 2 examples of the process of finding cited methodological information for each article. S3 Fig contains illustrations for all 15 articles in the case series. In many cases, reviewers were able to locate additional information. However, this study revealed 5 types of issues that arise when readers follow shortcut citations in search of detailed methods (summarized in Fig 6). The first problem was an inability to identify the cited material due to incomplete or inaccurate citations (e.g., incorrect author names, years, or DOIs) or dead website links. In some cases, reviewers could not find evidence that the cited source existed. The second problem was accessing the cited source. PDFs for some older articles were difficult or impossible to obtain. Many articles, supplemental files, and books were not open access. Subscriptions vary among institutions, and even scientists from well-funded institutions may have difficulty accessing paywalled articles. The third problem was that the cited method was difficult or impossible to find. Some textbook citations failed to provide specific chapters or pages, making it difficult to locate the cited content. In one case, the cited source was in a different language. The fourth problem was that the cited method was not adequately described. Examples included cited sources that did not describe the method or provided the same details as the citing paper. Authors sometimes cited older resources as shortcuts, raising concerns that the cited resource may not have accurately reflected the modern methods used in the citing study. The fifth problem was that in many cases, reviewers had to follow a chain of shortcut citations to locate a complete methodological description. Following a citation chain is time-consuming and readers may have difficulty identifying and accessing the cited material at each step in the chain.

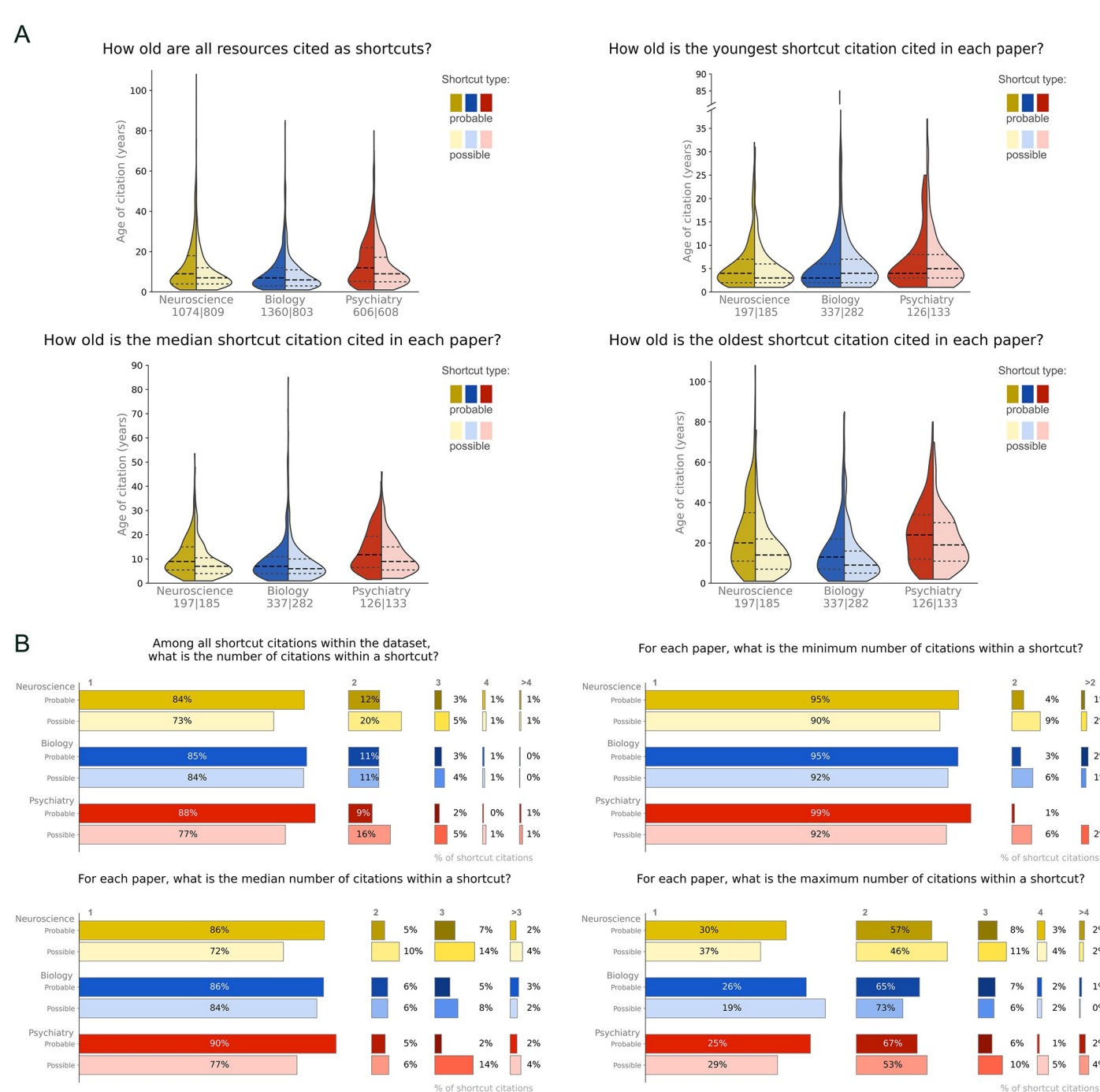

**Fig 3. Age and number of resources cited within a shortcut citations.** (A) The first violin plot shows the age of all resources cited as shortcuts, whereas the remaining box plots show the age of the youngest, median, and oldest resource cited as a shortcut for each article. The left side of the violin plot shows data for possible shortcut citations, whereas the right side shows data for probable citations. The youngest and median shortcut citation cited in a paper is typically less than 10 years old. The age of the oldest shortcut citation is more variable, and many of these citations were published more than a decade ago. The line with long dashes represents the median, whereas the lines with short dashes represent the 25th and 75th percentiles. The number of papers appears below the field name (*n* for probable shortcut citations/*n* for possible shortcut citations). (B) The first bar graph shows the number of resources cited within all shortcut citations in the dataset, whereas the next 3 bar graphs show the minimum, median, and maximum number of resources cited as shortcuts within an article. Percentages may not total 100% due to rounding errors. Data are available at https://osf.io/d2sa3/, in the methodological citations study folder [12].

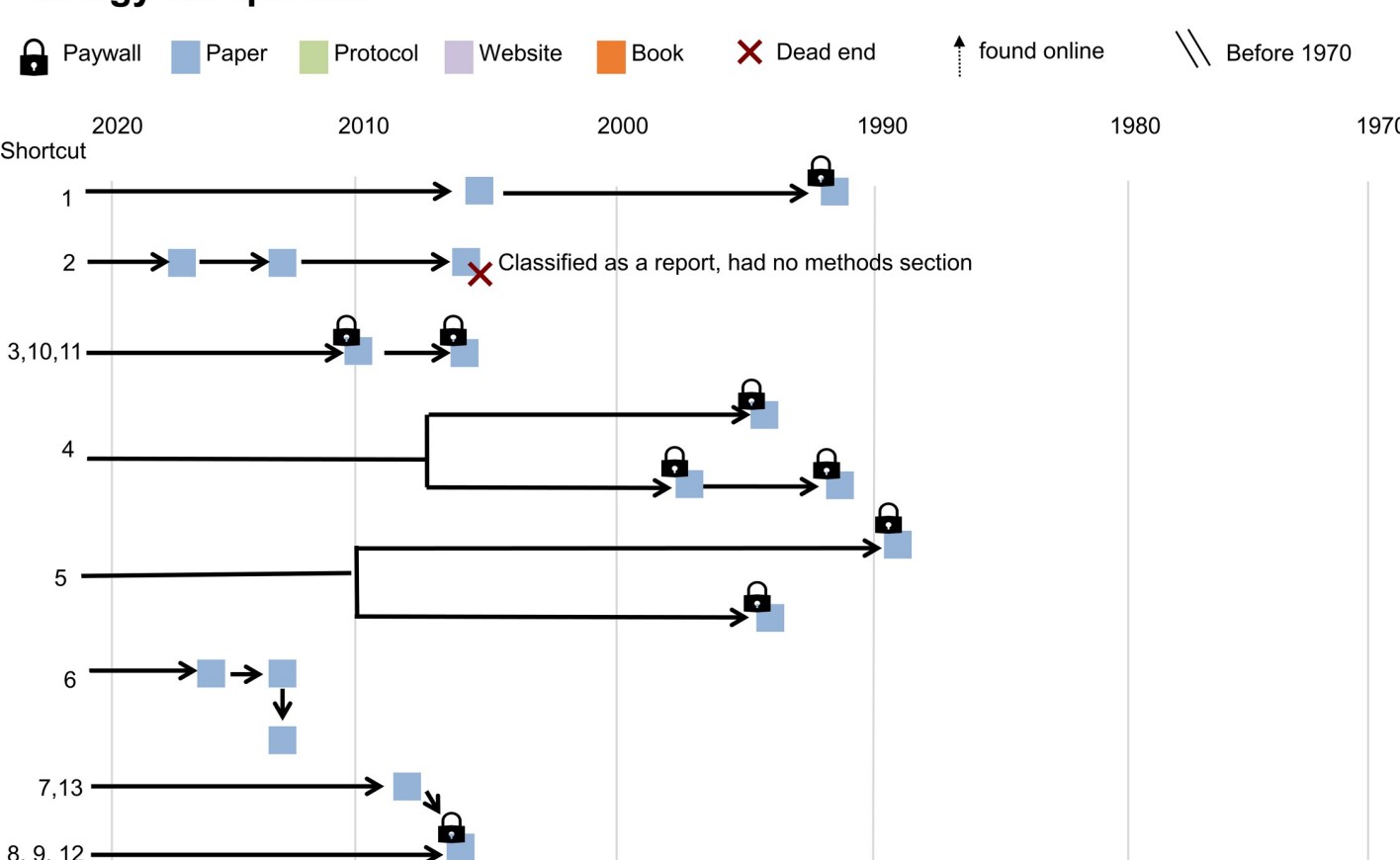

**Fig 4. Shortcut citations in example paper 1.** This diagram maps the process of finding methodological details for a biology paper in the fifth quintile of probable + possible shortcut citations in the shortcut citation chains study. The diagram shows the publication year and type of each cited resource and whether the resource was behind a paywall. Text on the diagram provides information describes problems encountered when searching for details about the cited method.

### Journal policy

**Study sample.** Among the 519 journals screened, 465 were eligible for the study (S4 Fig; 244 neuroscience journals; 76 biology journals; 145 psychiatry journals). Twenty-one journals appeared on the neuroscience and psychiatry lists and were included in both groups.

**Policies on details needed to reproduce experiments.** Policies that explicitly instruct authors to provide sufficient methodological details to allow others to reproduce the experiment were found in 40% of neuroscience journals, 18% of psychiatry journals, and 44% of biology journals (Fig 7A).

**Reporting of previously described methods.** Most journals had no policies concerning the reporting of methods that have been described previously (72% to 87%, Fig 7B). Some journals asked authors to summarize the method (9% to 19%) or to provide a citation instead of describing previously published methods in detail (4% to 8%). Policies asking authors to fully describe previously published methods were rare (0% to 2%). Policies asking authors to report modifications of previously described methods were also uncommon (10% to 21%, Fig 7C).

**Where to share detailed methods.** The percentage of journals that encouraged authors to share detailed methods in protocol repositories ranged between 12% and 23% (Fig 7D). A

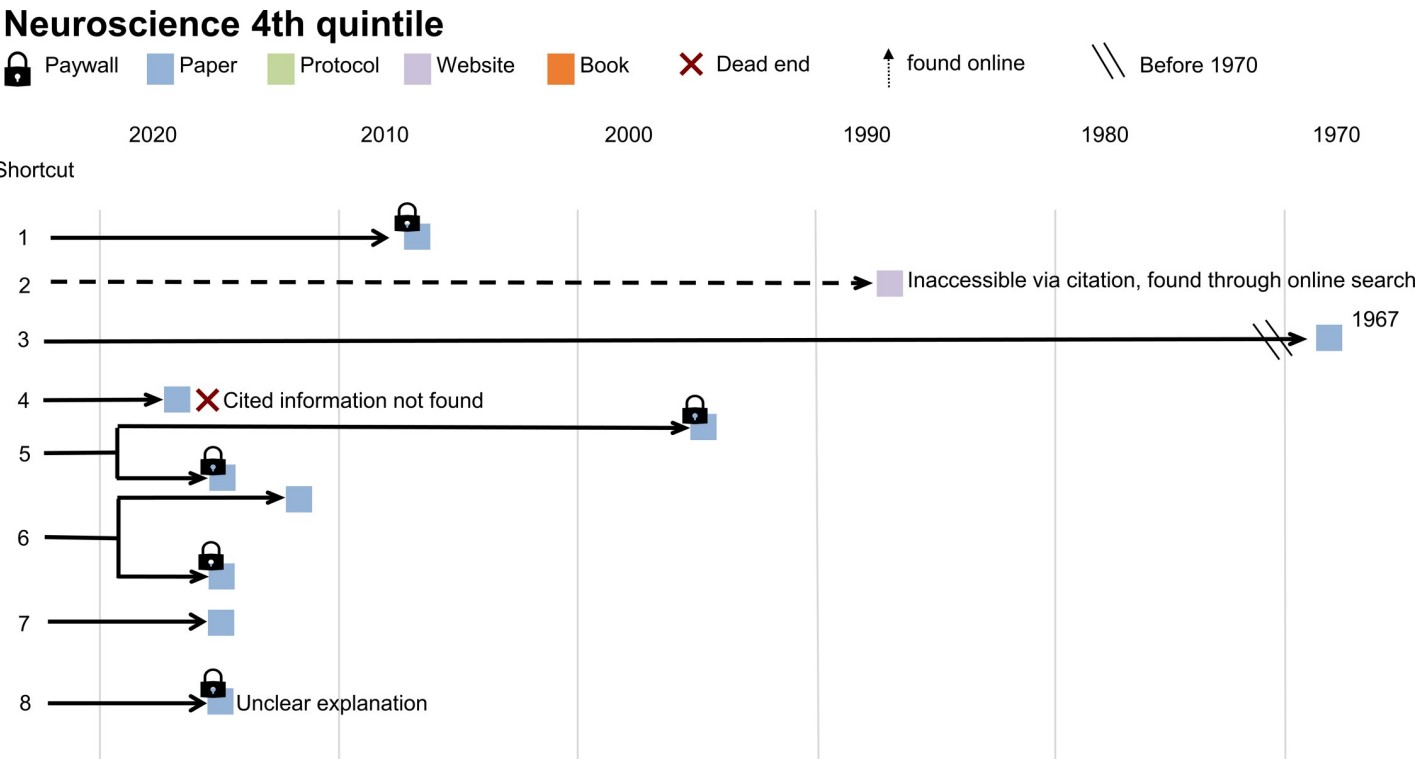

**Fig 5. Shortcut citations in example paper 2.** This diagram maps the process of finding methodological details for a neuroscience paper in the fourth quintile of probable + possible shortcut citations in the shortcut citation chains study. The diagram shows the publication year and type of each cited resource and whether the resource was behind a paywall. Text on the diagram provides information, describes problems encountered when searching for details about the cited method.

similar proportion of journals (9% to 21%) encouraged authors to share detailed methods elsewhere, without specifying where methods should be shared. Policies encouraging authors to share detailed methods in a protocol journal (0% to 8%) or as supplemental files (2% to 5%) were rare.

## Discussion

This exploratory study of papers in biology, neuroscience, and psychiatry revealed several important findings. First, citations are often used in methods sections. More than 90% of papers used a shortcut citation, explaining how a method was performed. Other common reasons for using citations in the methods included "who or what" citations, that give credit to the authors of another paper or specify what was used, and "why" citations, which provide context or a justification. Different methods evolve at different rates; however, citation age assessments suggested that some methods described in shortcut citations may no longer reflect current practice. The shortcut citation chains study showed that while shortcut citations can be used effectively, they can also create problems for readers seeking detailed methods. These problems included difficulty identifying the correct citation, accessing the cited materials, finding the cited method within the cited materials, and insufficient descriptions of the cited method. Following chains of shortcut citations to find methodological details was time-consuming, and each additional step in the chain can amplify the problems described in the previous sentence. Journals typically lack policies addressing methodological reporting. Fewer than one quarter of journals had policies addressing how authors should report methods that have been described previously, or address modifications of previously described methods. While some journals (18% to 43%, depending

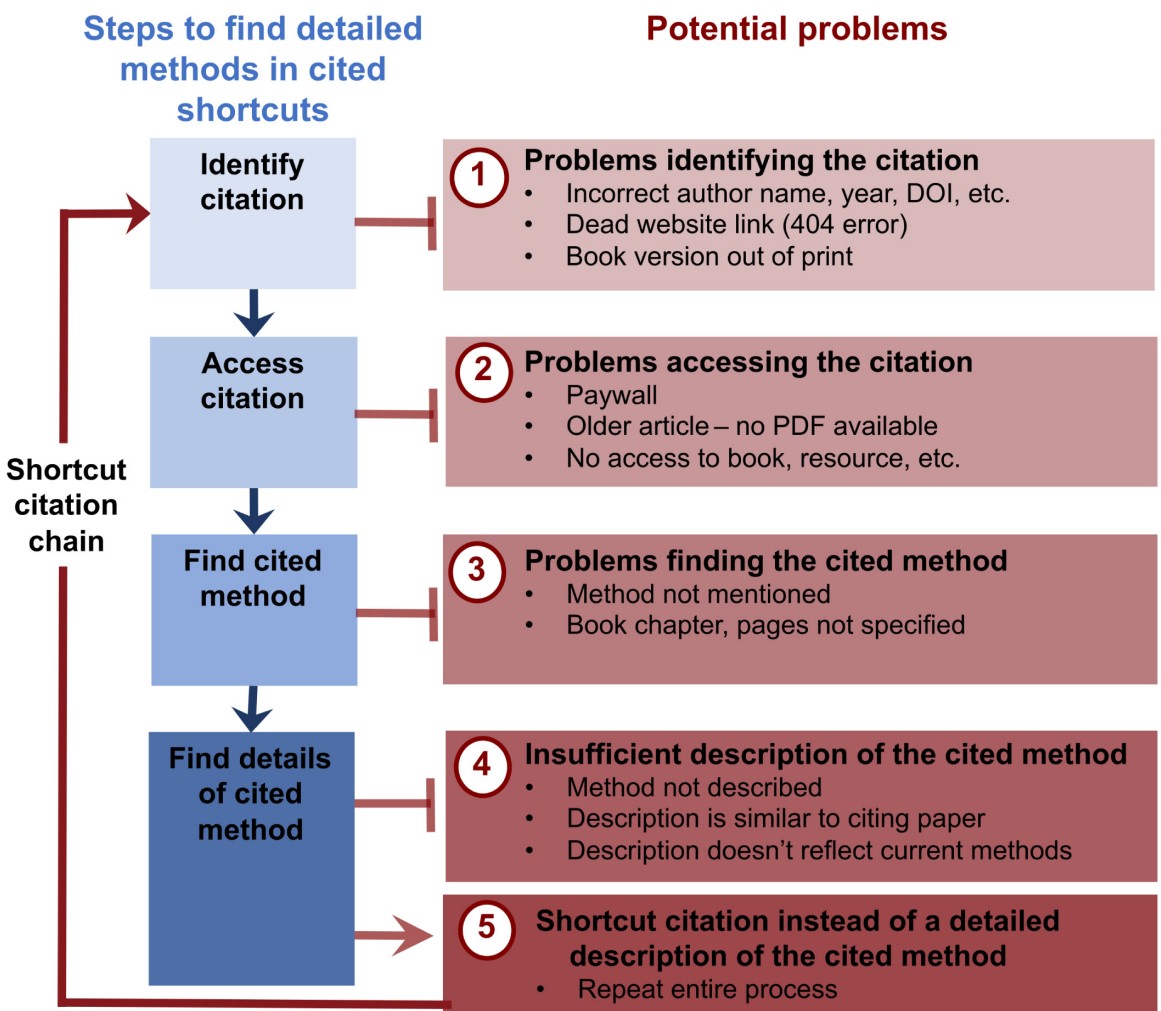

**Fig 6. Problems that arose when searching shortcut citations for detailed methods.** While methodological shortcut citations can be used effectively, reviewers encountered some problems when consulting shortcut citations to find details of cited methods. These included problems identifying the citation, problems accessing the citation, problems finding the cited method within the shortcut citation, and an insufficient description of the cited method. Chains of shortcut citations, in which the cited shortcut citation also used a shortcut citation to describe the cited method, were common.

on the field) asked authors to provide sufficient methodological details to allow others to reproduce the method, most journals (57% to 81%) had no such policy.

## Using shortcut citations to foster a culture of open and reproducible methods

When using shortcut citations, authors replace a section of their methods with a citation referring to another resource. The details contained within that resource are essential to implement the method. We therefore propose that methodological shortcut citations should meet higher standards than other types of citations. Box 1 outlines 3 proposed criteria that authors can use to determine whether a resource should be cited as a shortcut. The open access criterion may be controversial for some, as it suggests that scientists who have a paywalled resource that meets the other 2 criteria should cite this resource to give credit, and create a second, open

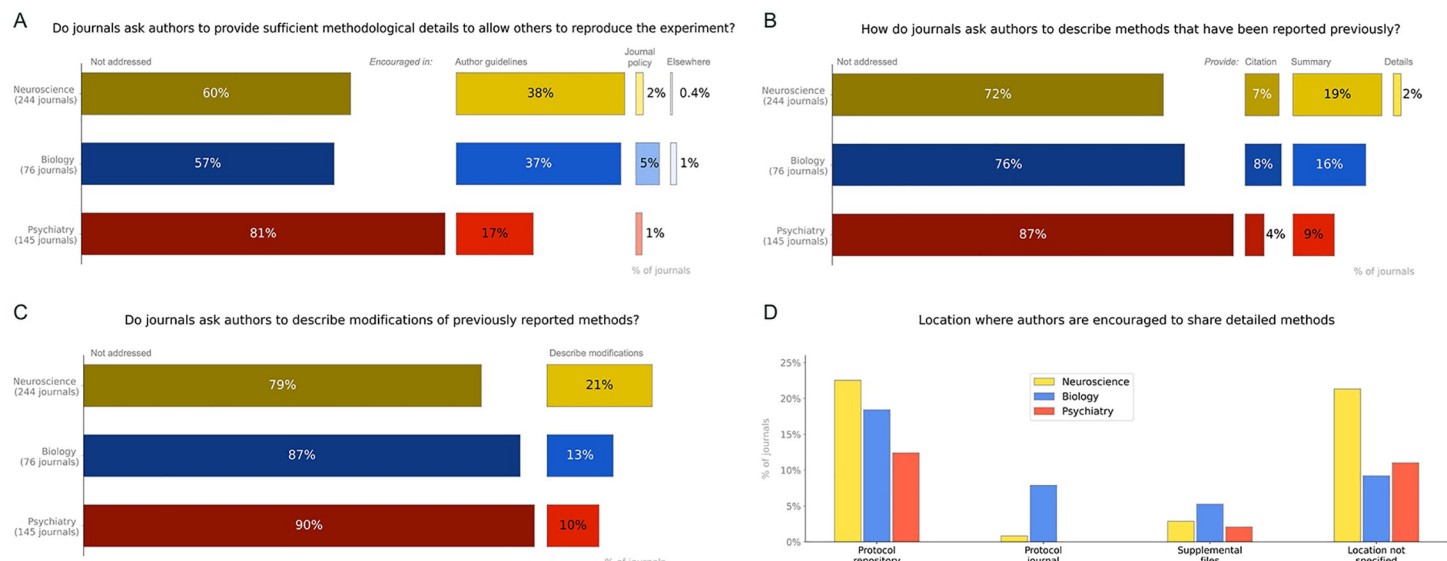

**Fig 7. Journal policies for methodological reporting.** Policies for all journals in 3 fields were assessed to determine (A) whether journals ask authors to provide sufficient information about the methods to allow others to reproduce the experiments; (B) how journals ask authors to describe methods that have been reported previously; (C) whether journals ask authors to describe modifications of previously reported methods; and (D) where journals ask authors to share detailed methods. Journals were represented in more than one category of panel D if they encouraged authors to share detailed methods and protocols in more than one place. Percentages in panels A–C may not total 100% due to rounding errors. Data are available at https://osf.io/d2sa3/, in the journal policy study folder [12].

access resource (e.g., a deposited protocol) to cite as a shortcut. Nevertheless, we believe that the open access criteria is particularly important. Unlike other types of citations, readers who want to implement the study methods will need to read shortcut citations. Paywalled shortcut citations systematically deprive some scientists of information needed to reproduce experiments.

---

### Box 1. How to use methodological shortcut citations responsibly

Methodological shortcut citations replace a section of the methods. Detailed methods are essential for those seeking to implement the method. We therefore propose that resources cited as methodological shortcuts should meet higher standards than materials cited for other purposes.

We propose that authors should use 3 criteria to determine whether a paper, or another resource, should be used as a shortcut citation. Resources that do not meet these criteria can be cited to give credit to those who developed the method, but should not be used as shortcuts.

1. ***Detailed description:*** *The resource should provide enough detail about the method that was used to allow others, including those who have little prior experience with the method, to implement the method. Resources with sufficient detail to be shortcut citations might include protocols, methods papers that are recent enough to reflect the current practices, or original research articles with unusually detailed methods.*

2. ***Similar or identical method:*** *The method described in the resource should be similar or identical to the method used by the authors. The authors should be able to describe any modifications to the methods in the methods section of their paper.*

3. ***Open access:*** *Paywalled or inaccessible shortcut citations deprive some readers of the information needed to implement the method. This creates disproportionate obstacles for researchers with limited access to publications, including researchers in countries with limited research funding and scientists who are not affiliated with a major institution.*

Authors using shortcut citations should:

1. ***Describe modifications in detail:*** *Deviations from the published method should be clearly described, in enough detail to allow others to implement the method. When a protocol posted on a repository is cited as a shortcut citation, the authors can version or fork the protocol to share their exact methods. Versioning is sharing an updated version of one's own protocol, whereas forking is sharing an adapted version of a protocol shared by others.*

2. ***Specify the exact location of the cited methods:*** *This might include providing page numbers where the cited method was described for books and manuals or describing the method name and location in the cited resource.*

When a resource does not meet the criteria proposed in Box 1, we recommend that authors cite the resource to give credit to its authors and use other strategies to share detailed methods. Options for sharing detailed methods include supplemental files, protocol repositories, and protocol journals (S6 Table). Authors who deposit or publish protocols can cite these resources as shortcut citations.

Sharing methods in supplemental files is suboptimal for several reasons. First, readers who have access to the paper may not have access to the supplement. While completing this research, we noticed that some publishers and journals make supplemental files freely available, whereas others do not. Papers obtained through online repositories or interlibrary loan programs may not include supplements. Second, methods in supplemental files are not findable. While scientists can quickly search protocol repositories to identify relevant protocols, there is no way to identify papers that contain detailed methods in the supplemental files. Third, supplemental methods are often written as general descriptions, which are typically less useful than the detailed, step-by-step protocols shared in protocol repositories and methods journals. Fourth, supplemental methods cannot be updated after publication; hence, this format is not useful for tracking the evolution of protocols within and across research groups. This is not a problem when investigators' goal is to describe their methods for a completed study. Methods, however, are constantly evolving and are one of the most useful outputs that researchers create. Given this, it is more efficient for research teams to use platforms that allow versioning and forking when sharing research procedures, and cite the version of the method that was used for a particular research study.

A key advantage of protocol journals is that they allow authors to obtain credit for their methods development work through a peer-reviewed publication. One disadvantage is that the publication process takes time and effort. Furthermore, the published method cannot be updated—it only shows what one research group is doing at a single point in time. In contrast, protocol repositories allow authors to create living protocols by quickly sharing updated versions [21]. Some repositories also allow forking, in which scientists can post a modified version of a protocol posted by someone else [21]. Versions and forks are linked to the original protocol. This credits the original authors for their work, while allowing researchers to see how the

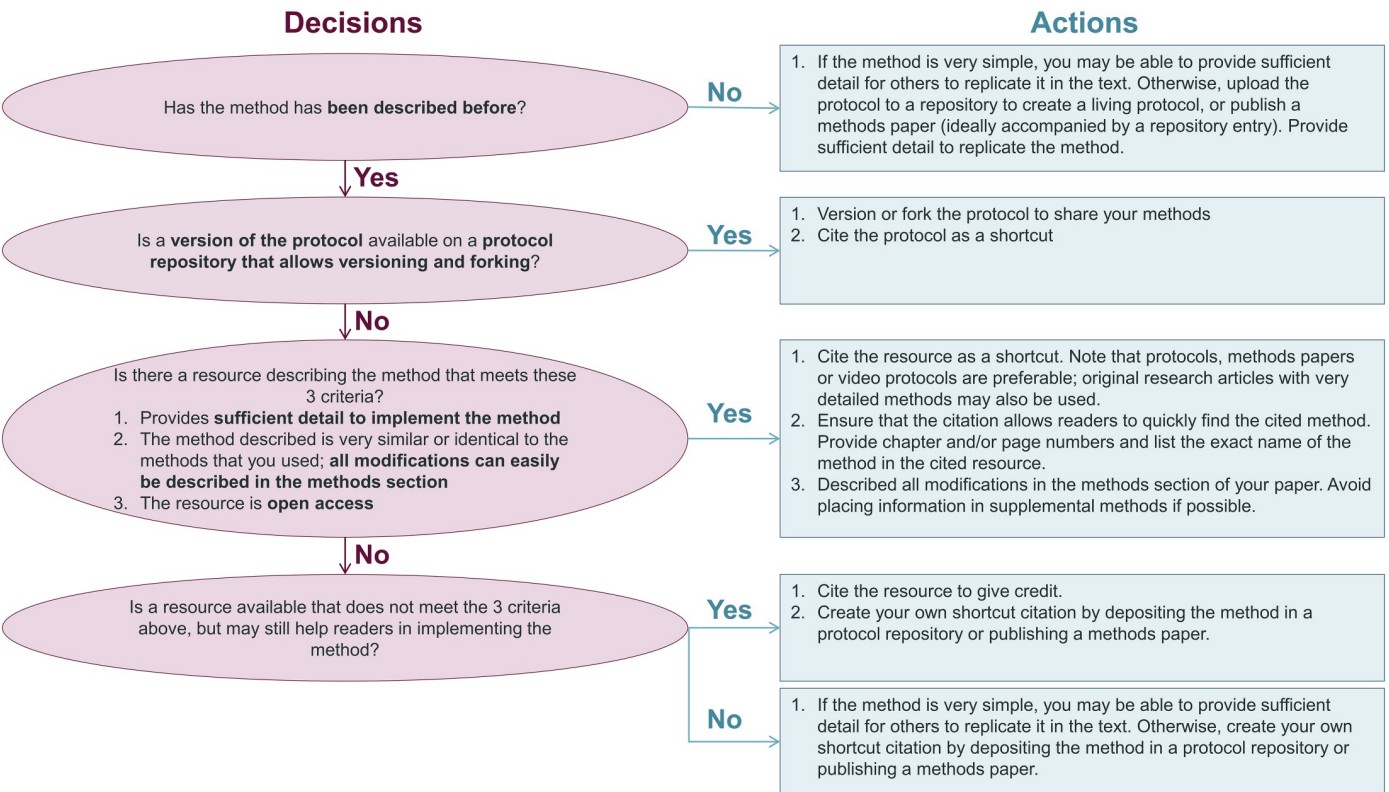

**Fig 8. Decision tree for the responsible use of shortcut citations.** This decision tree helps authors to prepare reproducible methods sections by determining when to use shortcut citations and when to share detailed methods through protocol repositories or methods articles.

protocol evolved. *PLOS ONE* offers a new "Lab protocol" publication [22] that combines the advantages of both formats. Lab protocol publications consist of a protocol, deposited on protocols.io [21], paired with a brief peer-reviewed publication that provides context for the protocol. Authors can demonstrate that the protocol works by citing previous publications that used the protocol, or providing new data obtained using the protocol.

The decision tree shown in Fig 8 may aid scientists in using shortcut citations responsibly, while reporting methods more transparently. Achieving these goals requires a shift in incentives —the scientific community needs to value protocols as a product of scientific work, on par with publications. While depositing methods takes time, it benefits authors in the long term. Depositing protocols in a repository that allows versioning and forking allows researchers to track changes as the protocol evolves and determine what version of a protocol was used for a particular publication. Furthermore, repositories provide long-term access to protocols even when researchers have not used the protocol in years, have moved to another lab or institution, or the person responsible for the protocol has left the research group. Detailed deposited protocols can also make it easier for new team members to learn protocols. Finally, other scientists can use and cite deposited protocols. This may facilitate collaborations with others who are building on one's methods or help to identify previously unknown factors that affect protocol outcomes.

While some scientists believe that one should always cite the first paper to use a method, others prefer to cite recent papers that fully describe methods that are similar to their own. These 2 beliefs reflect different reasons for citing a paper. Scientists who cite the original paper are using a "who or what" citation to give the authors of the cited paper credit for developing

the method. In contrast, those who cite recent papers that describe current protocols recognize that these citations are more useful as shortcut citations. Fortunately, authors can do both. Authors should structure the citing sentence to help readers to distinguish between the "who or what" citation and the "shortcut" citation. For example, authors might write: *"Experiments were performed using an updated version [citation 1] of the protocol originally developed by Smith and colleagues [citation 2]."* This phrasing clearly demonstrates that the first citation is the shortcut citation, while the second citation is intended to give credit.

## The role of journals in fostering open and reproducible methods

The scientific community has rapidly expanding groups focused on open data and open code [23–26], yet comparatively little attention has been dedicated to open methods. Open data are most useful when they are generated through rigorous experimental designs, using reproducible methods. Data are of little value when we do not understand or cannot evaluate the quality of the methods used to generate them. By implementing the actions outlined below, journals can foster a culture of protocol sharing, and open and reproducible methods, within the scientific communities that they serve.

- **Make all methods and protocol publications open access. When publishing paywalled papers, move methods sections in front of the paywall [27].** Methods and protocol papers are extremely likely to be cited as shortcut citations, and readers need to access these resources to determine how a method was performed. Eliminating paywalls for methods and protocol papers would allow everyone to access the information needed to implement the methods described. Moving methods sections for all closed access papers, including historical content, in front of the paywall, would give everyone access methodological details from papers cited as shortcuts. Publishers already place other parts of the paper, such as abstracts and references, in front of the paywall.

- **Replace or augment static methods and protocol papers with dynamic formats where authors publish a living protocol on an open access protocol repository:** The static formats used to publish traditional research papers, which are primarily focused on results, do not work well for methods. Protocols evolve over time. The question is not whether protocols will change, but when and how they will change. Furthermore, step-by-step protocols are often more useful that the text descriptions found in methods section. Journals should embrace dynamic formats for publishing methods, such as the *PLOS ONE* Lab Protocols article [22]. Authors publish a detailed living protocol on an open access protocol repository [21], which can be versioned and forked to track the evolution of methods within and across research teams.

- **Ensure that the protocol repositories used to publish dynamic methods allow versioning and forking:** Versioning and forking allow readers to share and track the evolution of methods over time, both within and across labs. Forking benefits protocol creators by giving them credit for their work, while allowing them to see how others are adapting their methods. Without forking, researchers may be reluctant to share adapted versions of protocols created by others, as they do not want to inadvertently claim credit for work that was not their own.

- **Exempt methods sections from word count limits. Ask authors to provide sufficient details to allow others to implement the method.** Word count exemption policies (e.g., [28]) eliminate a barrier to sharing detailed methods.

- **Encourage authors to share methods in protocol repositories, not as supplemental files.** Protocols deposited in repositories are findable and dynamic, whereas methods deposited in supplemental files are not.

- **Require authors use methodological citation shortcuts responsibly, according to the criteria described in Box 1. Eliminate policies requiring authors to cite previous publications instead of fully describing methods.** The research community should rethink the way that shortcut citations are used to ensure that these citations contribute to a culture of open and reproducible methods reporting. Policies that ask authors to use shortcut citations irresponsibly deprive others of details needed to implement the method. The driving force behind some of these policies may be concerns about plagiarism and copyright violations [5]. Laws in some countries specifically exempt detailed descriptions of protocols and methods from copyright restrictions (e.g., United States [29,30], European Union [31]), whereas other countries take a more indirect approach by not including detailed descriptions of methods under items that can be copyrighted (e.g., United Kingdom [32]). Other countries may not offer similar protection. Resolving these legal issues is crucial to allow scientists to fully describe their methods whenever they publish their work.

## Limitations

This study has several important limitations. Data examining the reasons for citing a paper in the methods (Fig 2A) should be viewed as a rough approximation. Abstractors were making distinctions about the reasons for a citation that the authors themselves likely did not make when inserting citations. Small changes in the wording or position of the reference could alter the categorization, as could variations in reader expertise. We may have systematically underestimated the number of shortcut citations, as we did not have the resources to assess all citations in supplemental methods files. When abstractors encountered a citation that could potentially fall under multiple categories, they were instructed to choose the most likely category. Abstractors encountered many citations that could have been classified under more than one category. Researchers who use this protocol in the future may wish to allow abstractors to select more than one category for each citation. Probable and possible shortcuts were defined using syntactic definitions. Conceptual definitions were not feasible, as reviewers' familiarity with the methods described was highly variable. Our results may not apply to journals with lower impact factors or non-English language journals. The shortcut citation chains study was designed to determine what problems one might encounter when following shortcut citations to find detailed methods. Data from this small, nonrepresentative sample (1 paper per quintile of possible plus probable shortcut citations per field) should not be used to determine how often each of these problems occurred. Larger samples would be needed to answer this question.

This study focused on shortcut citations. We did not examine the completeness of methodological reporting when authors described a method without using a shortcut citation. The 3 fields examined are within the life sciences and results may not be generalizable to other domains.

## Conclusions

Authors routinely used methodological shortcut citations to explain their research methods. While shortcut citations can be used effectively, they can also make it difficult for readers to find methodological details that would be needed to implement the method. Journals often lack clear policies to encourage open and reproducible methods reporting. We propose that methodological shortcut citations should meet 3 criteria. Cited resources should describe a method similar to the one used by the authors, provide enough detail to allow others to implement the method, and be open access. We outline additional steps that authors can take to use

methodological shortcut citations responsibly, and also propose actions that journals can take to foster a culture of open and reproducible methods reporting. Future studies should explore strategies for creating and sharing detailed protocols that others can use. The scientific community should also identify opportunities to incentivize and reward open methods.

## Supporting information

**S1 Fig. Methodological citations study flow chart.** This flow chart illustrates the journal and article screening process and shows the number of observations excluded and reasons for exclusion at each phase of screening. Data are available at https://osf.io/d2sa3/, in the methodological citations study folder [12].
(TIF)

**S2 Fig. Reasons for citing a resource in the methods section of a paper.** The box plots illustrate the number of times that each type of citation was used, per article, for neuroscience (yellow), biology (blue), and psychiatry (red). The horizontal line within each box shows the median, whereas the top and bottom of the box show the 25th and 75th percentiles. Whiskers represent the furthest datapoint that is within 1.5* the interquartile range from the box. Dots above the whiskers show outliers. Data are available at https://osf.io/d2sa3/, in the methodological citations study folder [12].
(TIF)

**S3 Fig. Diagrams for individual articles in the shortcut citation chains study.** These diagrams map the process of finding methodological details for each of the 15 papers in the shortcut citation chains study. Reviewers consulted resources cited in shortcut citations to find methodological details. The diagrams show the publication year and type of each cited resource and whether the resource was behind a paywall. Chains of shortcut citations occur when the cited source also uses a methodological shortcut citation to describe the method. Text on the diagram provides information describes problems encountered when searching for details about the cited method.
(PDF)

**S4 Fig. Flow diagram for journal policy study.** This flow chart illustrates the journal screening process and shows the number of observations excluded and reasons for exclusion at each phase of screening. Data are available at https://osf.io/d2sa3/, in the journal policy study folder [12].
(TIF)

**S1 Table. Number of articles examined for each neuroscience journal.** Values are *n*, or *n* (% of all articles). Screening was performed to exclude articles that were not full-length original research articles (e.g., reviews, editorials, perspectives, commentaries, letters to the editor, short communications), were not published in March 2020, or did not have a methods section. No issue indicates that the journal did not publish an issue or any articles in March 2020. Data are available at https://osf.io/d2sa3/, in the methodological citations study folder [12]. * Journals were included on both the neuroscience and psychiatry (S3 Table) lists.
(DOCX)

**S2 Table. Number of articles examined for each biology journal.** Values are *n*, or *n* (% of all articles). Screening was performed to exclude articles that were not full-length original research articles (e.g., reviews, editorials, perspectives, commentaries, letters to the editor, short communications), were not published in March 2020, or did not have a methods section. No issue indicates that the journal did not publish an issue or any articles in March 2020. Data

are available at https://osf.io/d2sa3/, in the methodological citations study folder [12].
(DOCX)

**S3 Table. Number of articles examined for each psychiatry journal.** Values are *n*, or *n* (% of all articles). Screening was performed to exclude articles that were not full-length original research articles (e.g., reviews, editorials, perspectives, commentaries, letters to the editor, short communications), were not published in March 2020, or did not have a methods section. No issue indicates that the journal did not publish an issue or any articles in March 2020. Data are available at https://osf.io/d2sa3/, in the methodological citations study folder [12]. * Journals were included on both the neuroscience and psychiatry (S1 Table) lists.
(DOCX)

**S4 Table. Methods repositories used in neuroscience, biology, and psychiatry.** Values are *n* (% of articles). Data are available at https://osf.io/d2sa3/, in the methodological citations study folder [12].
(DOCX)

**S5 Table. Age of the youngest, median, and oldest shortcut citations within a paper for each field.** This table shows summary statistics for Fig 3A. IQR, interquartile range. Data are available at https://osf.io/d2sa3/, in the methodological citations study folder [12].
(DOCX)

**S6 Table. Comparing methods of sharing detailed protocols.** JOVE, Journal of Visualized Experiments; N/A, not applicable; OA, open access. * protocols.io offers a partnership with *PLOS ONE* where authors can publish a methods article linked to their protocol.
(DOCX)

## Acknowledgments

The authors thank Małgorzata Anna Gazda and Alberto Antonietti for their assistance in translating policies of Polish and Italian journals.

## Author Contributions

**Conceptualization:** Britta R. Lewke, Kathleen Bastian, Anna-Delia Knipper, Orestis Rakitzis, Philipp van Kronenberg Till, Tracey L. Weissgerber.

**Data curation:** Kai Standvoss, Vartan Kazezian, Isa Steinecker.

**Formal analysis:** Kai Standvoss, Nina Nitzan Soto.

**Funding acquisition:** Tracey L. Weissgerber.

**Investigation:** Kai Standvoss, Vartan Kazezian, Britta R. Lewke, Kathleen Bastian, Shambhavi Chidambaram, Subhi Arafat, Ubai Alsharif, Ana Herrera-Melendez, Anna-Delia Knipper, Bruna M. S. Seco, Nina Nitzan Soto, Orestis Rakitzis, Isa Steinecker, Philipp van Kronenberg Till, Fereshteh Zarebidaki, Parya Abbasi.

**Methodology:** Vartan Kazezian, Britta R. Lewke, Kathleen Bastian, Anna-Delia Knipper, Orestis Rakitzis, Philipp van Kronenberg Till, Tracey L. Weissgerber.

**Project administration:** Kai Standvoss, Vartan Kazezian, Britta R. Lewke, Tracey L. Weissgerber.

**Software:** Britta R. Lewke.

**Supervision:** Vartan Kazezian, Tracey L. Weissgerber.

**Validation:** Kai Standvoss, Vartan Kazezian, Britta R. Lewke, Isa Steinecker, Tracey L. Weissgerber.

**Visualization:** Kai Standvoss, Vartan Kazezian, Shambhavi Chidambaram, Ana Herrera-Melendez, Anna-Delia Knipper, Nina Nitzan Soto, Orestis Rakitzis, Isa Steinecker, Philipp van Kronenberg Till, Fereshteh Zarebidaki, Tracey L. Weissgerber.

**Writing – original draft:** Vartan Kazezian, Britta R. Lewke, Kathleen Bastian, Shambhavi Chidambaram, Anna-Delia Knipper, Bruna M. S. Seco, Isa Steinecker, Tracey L. Weissgerber.

**Writing – review & editing:** Kai Standvoss, Vartan Kazezian, Britta R. Lewke, Kathleen Bastian, Shambhavi Chidambaram, Subhi Arafat, Ubai Alsharif, Ana Herrera-Melendez, Anna-Delia Knipper, Bruna M. S. Seco, Nina Nitzan Soto, Orestis Rakitzis, Isa Steinecker, Philipp van Kronenberg Till, Fereshteh Zarebidaki, Parya Abbasi, Tracey L. Weissgerber.

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
