## [Editor Report · Decision Letter 0]

1 Feb 2023

Dear Tracey, 

Thank you for submitting your manuscript entitled "Taking shortcuts: Great for travel, but not for reproducible methods sections" for consideration as a Meta-Research Article by PLOS Biology.

Your manuscript has now been evaluated by the PLOS Biology editorial staff, as well as by an academic editor with relevant expertise, and I'm writing to let you know that we would like to send your submission out for external peer review.

Once your full submission is complete, your paper will undergo a series of checks in preparation for peer review. After your manuscript has passed the checks it will be sent out for review. To provide the metadata for your submission, please Login to Editorial Manager (https://www.editorialmanager.com/pbiology) within two working days, i.e. by Feb 03 2023 11:59PM.

Kind regards,

Roli

Roland Roberts, PhD

Senior Editor

PLOS Biology

rroberts@plos.org

---

## [Decision Letter · Decision Letter 1]

15 Mar 2023

Dear Tracey,

Thank you for your patience while your manuscript "Taking shortcuts: Great for travel, but not for reproducible methods sections" was peer-reviewed at PLOS Biology. It has now been evaluated by the PLOS Biology editors, an Academic Editor with relevant expertise, and by four independent reviewers.

In light of the reviews, which you will find at the end of this email, we would like to invite you to revise the work to thoroughly address the reviewers' reports.

You 'll see that reviewer #1 is positive, but thinks that the assessment of the scale of the problem is somewhat weak, and probably requires a more quantitative treatment (and larger sample size); he also questions the field categorisation, asks about types of protocol (and field-specific citation culture), and makes a number of helpful textual and presentational requests. Reviewer #2 is very positive, and only has textual requests (plus a suggestion to include an example from Fig S2 in the main text). Reviewers #3 and #4 are also very positive and only have textual requests.

IMPORTANT: Please note that, contrary to a recommendation from reviewer #3, we do not accept structured abstracts. Also, note reviewer #2's recommendation to incorporate some aspects of Fig S2 into the main manuscript. During cross-commenting, one of the other reviewers also strongly supported this recommendation. I looked at this Figure to see what the fuss was about, and I agree that it's fascinating and interestingly presented - I encourage you to include a few "case studies" as part of a new main Figure.

Given the extent of revision needed, we cannot make a decision about publication until we have seen the revised manuscript and your response to the reviewers' comments. Your revised manuscript is likely to be sent for further evaluation by all or a subset of the reviewers.

**IMPORTANT - SUBMITTING YOUR REVISION**

*Re-submission Checklist*

*Published Peer Review*

*PLOS Data Policy*

*Blot and Gel Data Policy*

Sincerely,

Roli

Roland Roberts, PhD

Senior Editor

PLOS Biology

rroberts@plos.org

REVIEWERS' COMMENTS:

Reviewer #1:

[identifies himself as Olavo B. Amaral]

Summary:

The manuscript addresses the question of "shortcut citations" in methods description. Although this problem is frequently mentioned in debates about methodological reproducibility, it is understudied and it's nice to see actual research about it.

The results contain three main sections, which study (a) the prevalence of various types of citations in the methods sections of articles in highly cited journals, including shortcut ones, (b) examples of what happens when shortcut citations are followed and (c) a review of journal policies. This is followed by a reasonably extensive discussion focused on (d) guidelines on how to use shortcut citations.

I generally agree that this is an interesting structure, as it (a) documents the phenomenon, (b) evaluates to what degree it represents a problem, (c) inquires what is being made to address it and (d) suggests additional measures. The weakest link in the chain, however, seems to be point (b) (i.e. measuring the impact of the problem), as I am not sure the case studies provided are enough to quantify this. I will try to make this clear in my major points below.

Main concern:

- While the numbers of articles and citations in the first section of the study are probably sufficient to provide an overview of the use of citations, the 15 articles included as case studies in the second section are not. The authors seem to acknowledge this limitation, as they refrain from making a quantitative synthesis of these articles. That said, this leads this section of the manuscript to fall short in accurately presenting the importance of the problem. 

Although I found the visualization for each case study provided in Fig. S2 interesting, I would doubt that most readers will really make the effort to go through each one of them, much less be able to synthesize the data in their own heads to reach meaningful conclusions. Thus, I would strongly recommend that the authors provide some kind of quantitative synthesis of the problem in this section (i.e. What percentage of shortcut citations can ultimately be traced to the original reference? What's the average number of steps? What percentage is behind a paywall? What percentage reaches a dead end or an insufficient description?).

I note that 15 articles are probably too few for this purpose, and that the sample of articles in which citations are followed would have to be expanded. Thus, I would recommend that the authors perform a sample size calculation to reach the number of citations/articles that can provide reliable estimates within a given confidence interval. For this purpose, it's worth noting that it would be desirable to perform synthesis both at the level of citations (i.e. what percentage of citations in the sample can be traced?) and at the level of articles (i.e. what percentage of articles in the sample have at least one untraceable citation?), as citations within a single article should not be considered as fully independent units when it comes to representing the whole population of citations. Thus, using articles as units for the purpose of sample size calculation might be the better option.

Other general concerns:

- The categorization of scientific fields is somewhat strange: most people would probably consider neuroscience is a subfield of biology, so presenting both as separate categories may puzzle some readers. I understand that this is a consequence of the JCR categories used, but making this clearer from the start (e.g. "examine the use of shortcut citations in neuroscience, biology and psychiatry journals in the abstract) and perhaps referring to the biology journals as "general biology" would help to avoid confusion.

Still on this point, the selection of fields is narrow and ad hoc. I understand that this is a limitation posed by the authors' own expertise, but it is nevertheless one of the main weaknesses of the manuscript. Thus, the narrow range of scientific fields examined should probably be mentioned in the limitations section.

- Even within this relatively narrow sample of fields, the kinds of methods that deserve a protocol probably varies a lot: I'd guess that psychiatry journals include a lot of surveys and instruments, while biology and neuroscience might have predominantly wet lab protocols. It would be interesting if somewhere in the paper (possibly in the example cases provided) we could get a feeling of what kind of "protocols" we are talking about, even if only in a general sense. If quantifying/classifying them is not feasible, at least some illustrative examples could be provided. Are we talking about methods to quantify proteins? Scales to measure depression? Electrophysiology setups for rodents)? The citation culture probably depends a lot on the particular method, so the whole discussion sounds a bit disembodied without touching on this point somewhere.

- Why are only minimum/maximum numbers of citations within shortcuts and the youngest/oldest citation coded? This looks like an approach to simplify data extraction, but it ends up providing very limited information (i.e. especially if there are many citations per paper, the oldest and youngest ones give very little information on the actual range).

Moreover, this ends up making data visualization in Fig. 3 much less intuitive than it could be (i.e. it would clearer and more informative to provide the full range of citation ages). If the authors could provide the full ranges (although I'm not sure that this is feasible), this would likely strengthen the paper. If not, I'd reconsider whether Fig. 3 should be included in the main results, as I don't think the results as displayed say much about the sample of citations as a whole.

- Some points in the case series description and discussion mention that some references "provided a description that was no longer state-of-the-art" and that this may be a problem. I don't really get the idea here: methods citation are supposed to provide an accurate description of what was done in a study, not of what's the current state of the art of the method. In this sense, descriptions shouldn't age badly or become "not-state-of-the art".

I understand the concern that a very old shortcut citation raises suspicions that it might not really describe what was done in the paper (as it may be likely that no one uses certain methods in exactly the same way after 50 years). But if this is what the authors meant, this should be stated more clearly, as it is not really the impression that comes out of reading these passages.

In the same vein, mentioning in the discussion that "supplemental methods cannot be updated" is technically correct, but is not a limitation in terms of making methods sections reproducible (which seems to be the point of the paper). For the purpose of methods description, whatever was used in a paper should remain static, even though the method may evolve in subsequent study.

- In terms of data sharing, one thing I could not find in the manuscript or in the OSF was the DOI and title of the articles used as case studies in Fig. S2. I may have missed it, but as there was no folder for the case series section I didn't know where to look for it. As this seems important for reproducing the findings, this list should be provided somewhere (possibly as a document within the OSF) and cited within the text and legend to figure S2.

Minor concerns:

Introduction:

- The correct name of the project mentioned in the first paragraph is Reproducibility Project: Cancer Biology (not "for Cancer Biology").

- "This risk of bias for randomization sequence generation and allocation concealment was unclear…" - this sentence seems odd (in particular the "This" at the start), please revise the wording.

Figure 1:

- Isn't the methods section a viable alternative for sharing details needed to reproduce experiments as well? While I agree that in many cases a separate protocol may be a better option, that depends on the length of detail that is needed, which will vary greatly depending on the method. Therefore, I would argue that the methods section should be included as an option in the figure - saying that the information "should" be shared in a separate document sounds overprescriptive.

- The second "readers" can be omitted from the third sentence of the figure legend.

Methods:

- Instead of citing the full OSF page for "protocols, data and code for the prevalence study and journal policy studies" using a single link, wouldn't it make sense to cite a specific DOI for each of these resources? The same thing hold for points in the text in which specific resources are cited (e.g. "The full search strategy is available on the OSF repository" could point to a direct link to the search strategy rather to the full OSF page).

I think this is optional, as the Readme files in the OSF are clear. But providing specific links to each resource would be more consistent with the authors' recommendation of providing pages for book citations, for example (in the sense of sparing the reader the trouble to search for a resource within a larger space).

- What is meant by "top journals" exactly? Are those the ones with the highest impact factor in the JCR in their specific fields? Although this would be my guess, it is not clear from the description.

- The data on whether papers were related to SARS-Cov2 sounded rather gratuitous, as Covid-19 was not mentioned anywhere in the introduction. If this data is to be kept in the paper (I personally don't think it adds much), the rationale for extracting this should be mentioned somewhere.

- Though this eventually became clear, I initially had a hard time to understand what was meant by "number of citations per shortcut". This could be made clearer when this variable is first introduced.

- The description of a probable shortcut states that "additional details are not provided in the following sentences or elsewhere in the methods sections". But what happens if the method is fully explained outside of the methods section (i.e. in the supplementary material or in a repository)? I was unsure how these cases were classified, so it's probably worth commenting explicitly on it.

- Electronic searches were performed using the terms "[journal name]", "journal citation reports ranking", "author guidelines", "journal policy", and "impact factor". I don't quite get what this search means to achieve. Why would one need to search for "impact factor" to look for policies?

Results:

Figure 2:

- The different areas have different mean numbers of methods citations per paper (being somewhat higher in Biology). Thus, showing the results for different categories in percentages as in Fig. 2A may cause misleading impressions - although there are still less "How" citations in Biology than in Neuroscience or Psychiatry when measuring absolute numbers, the actual difference is smaller (while that in "Who or what" citations is even larger). Having the bars represent absolute numbers (possibly still displaying the percentage within the bars) - with overall longer bars for Biology - would likely provide a more accurate impression of what's going on.

- It took me a while to understand the right panel in Fig. 2B. While the fact the two sides of the violin plots represent different data eventually becomes clear, wouldn't it make it easier on the reader to break the information for probable and possible citation into separate plots (especially as the left panel uses symmetric violin plots)?

Tables S5 and S6:

 - Can't the information in these tables be included in the legend for Fig.2 and Fig.3 (as it is relatively short and essentially synthesizes the data in the figures)? This is optional, but would leave the information in one place instead of creating a lot of supplements.

Figure 5:

- Are the categories in Fig.5A and 5D mutually exclusive? It would seem to me that a journal could encouragd providing sufficient methodological details both in the author guidelines and as policy, and that they may encourage sharing methods in more than one place (i.e. repository or supplemental files). This is likely worth commenting on in the legend.

Discussion:

- I don't think the Germany and California examples mentioned in Box 1 are needed: there are plenty of places of the world with much worse access, and these particular examples are not particularly representative of difficulties faced by the world at large.

- While I agree with the recommendation to "make all methods publications open access", I don't think that there's any particular reason why methods papers are different from the rest of science (in the sense that they should be open access), so I'm not sure the recommendation really belongs here.

- The discussion about copyright issues described in the list of recommendations is long for an item in a list. Thus, it probably would fit better in the main text or in a box.

Table S7:

- I get the feeling that Table S7 would read better if lines and columns were reversed (i.e. methods as lines, features as columns), but it may be a matter of taste.

- Why are supplemental files and protocol journals deemed static while shortcut citations are not? This does not make much sense to me.

- I'd say supplemental files would generally be expected to have been peer reviewed. I agree that this is likely not always the case, but that probably depends more on the reviewer than on the journal (e.g. I don't know of journals that explicitly exempts supplementary material from the peer review process), so I'd remove "depending on the journal". 

- The comment "protocols remain available over time" made for repositories stands for all categories - it makes sense when comparing a protocol repository to a lab notebook, not to the other forms of describing protocols. Thus, I'd probably not include it as an advantage here.

- I'd argue that both shortcut citations and supplemental files are "findable" for whoever's reading the paper (which is likely what matters here), so I'd be inclined to remove this category. 

- Clinical journals are not the only one to publish protocols as articles (the systematic review community has a tradition of publishing protocols, for example).

Figure 6:

- In the last no/no option, describing the method in the main text (if it is simple enough to fit) should also be included as an alternative. 

Reviewer #2:

[identifies herself as Emma Ganley]

Review of PBIOLOGY-D-23-00204R1, "Taking shortcuts: Great for travel, but not for reproducible methods sections".

------------------

Disclaimer/COI: 

------------------

As disclosed to the editors, to be fully transparent I'll include this information here: I currently work at https://protocols.io and therefore have a very obvious positive bias about the value and importance of protocols and methods sharing as a part of the scientific research process. I have also participated in some working groups with Tracey Weissgerber. And less relevant here but in the interests of full disclosure I was previously employed in an editorial capacity at PLOS Biology.

------------------

Overall View and Recommendation: 

------------------

This is an important meta-research study into the status of quo of how methods and protocols are cited in the scientific literature. The authors perform a detailed and thorough investigation of how researchers cite methods, the issues encountered by a reader who consults citations to understand methods that were used, and protocols followed, and how journal policies differ with respect to requirements or encouraged behaviour relating to methodological reporting.

This is important because methods sharing has lagged behind expectations around the sharing and availability of other research outputs like data, code, articles themselves etc. And yet insight into methods is critical for interpretation and reuse of data.

With Minor Revisions I would recommend this for publication in PLOS Biology. I don't have many comments and the first one is my subjective take on things, so will leave to editors/authors to decide whether to make changes or not in response.

------------------

Comments:

------------------

1. My one main comment is possibly a bit subjective, and the authors may have considered this already, but for the categories shown in Table 1 (for the reasons for citations in the methods sections), the difference between 'Who or What' and 'From where' isn't very clear. Both really seem to me to be ways to give credit: 

- for the 'Who or what' for the wording used is: "give credit to the group that created the method, tool, resource or substance, …" 

- and for the "From where" category the wording is that "The citation is used to show where a substance or organism was obtained from", and the example given is a lab as source of a mouse - this seems to be as much of a "Who" as a "From where".

Both categories include "substance" and seem to encompass resources/tools/materials - a mouse model could as easily fall into the Who or What as a Where from, I think these could easily be the same category. 

In the results the numbers for the "From where" are quite small, I wonder if these categories might be better off all rolled into one? I appreciate that this would involve a rework of how results are reported and some of the assessment, but worth thinking about whether these are distinct categories (you could still show sub-types within as you already do for the Who or what section in Fig 2).

2. Very minor: Under the description for Case Series on P9, it's noted that an assessment was made to determine if the methodological description was "adequate" - I can infer what is meant by adequate, but it might be helpful to add a definition of what would qualify as adequate.

3. Very minor pedantic comment: The order of Neuroscience / Psychiatry / Biology in the Key for Figure 2 is different from the order that the categories are shown in for each of the figure parts where they're always shown in the order Neuro / Bio / Psych. Suggest authors switch the key order to be the same as the figure panels.

4. Journal policy assessment is interesting to read through, but I wonder if the authors might care to suggest an ideal wording for a journal policy that could then be adopted by journals who want to do better. Even if worded as encouragement rather than mandate this might very valuable and appreciated (of course as with all these things it may also be contentious, so for authors to decide if they want to go there or not…). Another option could have been to identify if there were any exemplary journals out there already, with or without the wording that they use?

5. While I'm well versed in what forking means for a methods-sharing platform like protocols.io, personal experience has made it clear that to a lot of researchers the terminology of forking it not as clear as it could be. The authors should consider including a short definition/description of forking so that there is an easier understanding of the value of being able to easily fork a protocol (maybe using an example e.g. easy creation of a similar protocol where tweaks are needed in order to apply it a different organ or cell type, or to use a new piece of equipment, and how this is facilitated if you can start with a forked copy of an existing protocol so that protocol creation from scratch is not needed).

6. I really like the citation pathway representations in Fig S2 - is it worth putting one example into the main text for a reader to see one without having to go to the Sup file. I think it's very illustrative of the entire article.

Reviewer #3:

[identifies himself as Prof. Lucas Helal, PhD]

Dear authors, 

I congratulate you all for this very important piece of research I had the privilege to review. Please enface my comments as a way to improve your manuscript and not as a criticism. Also, I am going with some doubts throughout the text I hope you may have the opportunity to explain that in details. 

### Comments ###

1. Title: for my sense, the title should be more technically informative, specially in terms of design. Please use the Murad et al (2017) reporting guideline for meta-research studies to guide your writing.

2. Abstract: it is a matter of style, but for the vast majority of the times, a structured abstract works better for readers (and referees) than a non-structured one. I strongly recommend a structured abstract for your manuscript, although not being an exigence of PLOS Biology journal. 

3. Main text: 

3.1. Structure of sub-studies: 

- I'm up to believe your first study does not fit into a classification of a prevalence study (indeed, there is no prevalence study among study designs, but cross-sectional studies). To gather a true prevalence, your denominator should be all manuscripts of the studied population, which is not true. What you have in hands are relative frequencies. Therefore, for my sense, to claim fhe first study as a prevalence study is not correct at all. I would recommend "bibliometric analysis" OR "cross-sectional analysis". So, it serves also when presenting results - you cannot sustain prevalences in your report, rather than frequencies. It should be switched.

- As for your second study, I won't classify as a case series study as well. This is a clinical classification, although I understood the analogy. A term that is well accepted in the meta-research literature is a "cohort of studies", which I recommend for. 

- Finally, your third study also does not fit into your classification for my understanding. What is a "journal policy study"? When you are cross-sectionally analyzing journal policies (which you did, not longitudinally), it is a cross-sectional study, but your unit of analysis are not patients - rather, studies. Then, I strongly suggest to revisit it.

3.2 Stratification of studies

- It wasn't clear for me. Did you take all of your studies, split them into quintiles of their JCR, and randomly selected a fraction of studies of each quintile? If yes, it does make sense for me, but I would love to hear from you if you think that JCR may be a proxy of editorial policies (word limit) and quality of reporting; and, how the potential role of the study design and the nature of intervention were accounted for? I explain: RCTs and pre-clinical studies usually requires more detailed methods explanation than other designs; or, for a given discipline (Psychiatry, for example), a trial published to pursue an FDA/EMA market approval is completely different of a trial in which the intervention is physical activity, for example (potentially). This should be highly acknowledged - if not possible to control, acknowledge as a limitation in your discussion.

3.3 Results

- Personally, I think that is clearer for the readers to be redundant in the main body and figures/tables when talking about results. I felt lack of clarity in results, specially in the explanation of the violin plots. Is that possible for you to use redundancy and write the results presented in figures/tables as whole text as well? Quite an analogous of the methods shortcut. We can't presume that all readers do know to interpret three-factors (or more) plots easily and nowadays to make science communicable to peers is more imperative than ever.

- I liked the way you structured the logical of presenting results. They are suitable in a timeline manner and make sense intrinsically. 

3.4 Discussion

- I liked as well how you dedicated plenty of space for the future recommendations. We definitely should bring solutions rather than accuse the problems. This is easy. I congratulate you as a whole about this. My only minor point is that I felt some explanation how you determine "why" shortcut citations were used if a mixed-methods study was not conducted. Is it an inference (not statistical)? Did you gather any proxy? For my sense, this is a very difficult question to answer based only in a manuscript screening. 

4. Final Considerations

I appreciated this piece of research. Personally, in times in which the discussion of journal standards for submission and etc. is very pertinent. I advocate for formats like this, although I disclose I don't recognize it as a potential conflict of interest to review this manuscript. My central final question for you is: among your all recommendations, how to deal *pragmatically* with the problem of shortcut methods citations while culture changing doesn't finish your cycle (which takes a lot)? We do know that hybrid journals will be still reluctant to expand room for words; we do know that, still nowadays, researchers are afraid to deposit supplemental material and/or data (by the way, congratulations for the data-sharing practice) or even don't know how to do it if they want either - and the list goes on. Theoretically all recommendations make a lot of sense for me, but the problem seems to be urgent and a pragmatic solution needed. Do you have something in mind?

Reviewer #4:

[identifies himself as Timothy M Errington]

This paper reports an exploratory study looking at shortcut citations, an approach used to cite a previous paper instead of describing the methods in detail. The authors found that citation used in methods sections were common with methodological shortcuts common among the 3 fields investigated. Case studies were also conducted to examine how methodological information was able to be located (or not) for several papers. Finally, the paper examined the journal polices around making methods sections complete and reproducible. Overall, this paper highlights a major challenge in how scientific information is communicated, specifically around the methods used in published research findings. The paper is very detailed and clearly communicated. My suggestions below are recommendations to the authors to consider when revising their paper.

(1) While this paper focuses on the way authors use citations in methods for papers published during March 2020, do the authors have any way to help readers understand how this relates to how methods were reported in the past? That is, is this a new phenomenon or something that's been entrenched in scientific methods sections over the years? This is beyond the scope of the study, so my question is only if there are other papers on this topic that might be considered in the discussion. If not, it might be relevant to also discuss that it is unknown to the authors how these findings compare over time because of a lack of identified literature on the topic.

(2) Did the authors look at whether the short-cut citations were self-citing? That is, how common are short-cut citations to one's prior work?

(3) I believe the reason for why citations in supplemental method were not examined was because of a logistical constraint. The authors discussed their subjective impression that many supplemental methods lacked details or described specific details (e.g., list of primers). However, do the authors believe that shortcut citations also exist in supplemental methods and/or repositories? And if so, what potential impact would this have on how readers should interpret the findings reported in this paper? Is it a 'best case' scenario and that the number and incidence of shortcut citations is likely to be higher than what's reported here?

(4) While the authors did not look at completeness of reporting or other aspects of methods reporting, something that shortcut citations reminds me of is another practice that occurs in methods sections where information is not fully detailed, specifically using the phrase 'following standard protocol' instead of writing out or properly linking to the methods used, especially since what is 'standard' varies by lab. While this might be beyond the scope of this paper, the authors might consider how this practice is similar or not to shortcut citations and how this might impact the recommendations proposed.

(5) The authors might also consider this initiative (https://elifesciences.org/inside-elife/c40ae742/elife-latest-request-detailed-protocols-easily-via-bio-protocol-integration) in addition to the one from PLOS One highlighted in the discussion.

(6) Figure 6 has a typo in the bottom left circle - it should be 'that does not meet' instead of 'the does not meet'

---

## [Editor Report · Decision Letter 2]

8 Feb 2024

Dear Tracey,

Thank you for your patience while we considered your revised manuscript "Taking shortcuts: Identifying strategies for using methodological shortcut citations responsibly" for publication as a Meta-Research Article at PLOS Biology. This revised version of your manuscript has been evaluated by the PLOS Biology editors and the Academic Editor.

Based on our Academic Editor's assessment of your revision, we are likely to accept this manuscript for publication, provided you satisfactorily address the following data and other policy-related requests.

IMPORTANT - please attend to the following:

a) Please change your title to something more explicit and informative. We suggest "Shortcut citations in the methods section: prevalence, problems and strategies for responsible use," but I'm happy to discuss this via email.

b) I noticed when writing a blurb that some of your major findings (or at least that many potential readers will find to be striking, IMHO) do not appear in the Abstract, namely that (study 1) analysis of >800 papers across three fields shows that >90% of papers contain at least one shortcut citation, and that (study 2) these cause serious problems in understanding the method. I wonder if these could be more explicitly highlighted in the Abstract? (the results of study 3 do seem to be described)

c) Many thanks for providing the underlying data and code in OSF. Please could you cite the location of the data clearly in all relevant main and supplementary Figure legends (Figs 2ABC, 3AB, 7ABCD, S2, I think), e.g. “The data underlying this Figure can be found in https://osf.io/XXXXX”

We expect to receive your revised manuscript within two weeks. 

*Published Peer Review History*

*Press*

Sincerely,

Roli

Roland Roberts, PhD

Senior Editor

rroberts@plos.org

PLOS Biology

DATA NOT SHOWN?

---

## [Editor Report · Decision Letter 3]

26 Feb 2024

Dear Tracey,

Thank you for the submission of your revised Meta-Research Article "Shortcut citations in the methods section: frequency, problems and strategies for responsible reuse" for publication in PLOS Biology. On behalf of my colleagues and the Academic Editor, Cilene Lino de Oliveira, I'm pleased to say that we can in principle accept your manuscript for publication, provided you address any remaining formatting and reporting issues. These will be detailed in an email you should receive within 2-3 business days from our colleagues in the journal operations team; no action is required from you until then. Please note that we will not be able to formally accept your manuscript and schedule it for publication until you have completed any requested changes.

Sincerely, 

Roli

Senior Editor

PLOS Biology

rroberts@plos.org